# LONG-TERM TYPHOON TRAJECTORY PREDICTION: A PHYSICS-CONDITIONED APPROACH WITHOUT RE-ANALYSIS DATA

**Young-Jae Park** [*1]**, Minseok Seo**[*2]**, Doyi Kim**[2]**, Hyeri Kim**[2]**, Sanghoon Choi**[3]**, Beomkyu Choi**[2]**,
**Jeongwon Ryu**[2]**, Sohee Son**[2]**, Hae-Gon Jeon**[1]**, Yeji Choi**[2]
[1] Gwangju Institute of Science and Technology, [2] SI Analytics, [3] The University of Manchester

## ABSTRACT

In the face of escalating climate changes, typhoon intensities and their ensuing damage have surged. Accurate trajectory prediction is crucial for effective damage control. Traditional physics-based models, while comprehensive, are computationally intensive and rely heavily on the expertise of forecasters. Contemporary data-driven methods often rely on reanalysis data, which can be considered to be the closest to the true representation of weather conditions. However, reanalysis data is not produced in real-time and requires time for adjustment because prediction models are calibrated with observational data. This reanalysis data, such as ERA5, falls short in challenging real-world situations. Optimal preparedness necessitates predictions at least 72 hours in advance, beyond the capabilities of standard physics models. In response to these constraints, we present an approach that harnesses real-time Unified Model (UM) data, sidestepping the limitations of reanalysis data. Our model provides predictions at 6-hour intervals for up to 72 hours in advance and outperforms both state-of-the-art data-driven methods and numerical weather prediction models. In line with our efforts to mitigate adversities inflicted by typhoons, we release our preprocessed *PHYSICS TRACK* dataset, which includes ERA5 reanalysis data, typhoon best-track, and UM forecast data.

## 1 INTRODUCTION

As global warming accelerates, the intensity of typhoons is on the rise (Walsh et al., 2016; IPCC, 2023). Accurate typhoon trajectory prediction is of paramount importance to allow enough time for emergency management and to organize evacuation efforts. This warning period is more critical in countries that lack sufficient infrastructure to forecast typhoons. Many nations utilize numerical weather prediction (NWP) models for typhoon trajectory forecasting and base their typhoon preparedness measures on this forecasting information. NWP models make predictions for atmospheric variables like geopotential height, wind vectors, and temperature using atmospheric governing equations. Forecasters employ the specific characteristics of typhoons in their NWP output to infer the typhoon trajectory.

However, interpreting the outputs of NWP models depends on the expertise of the forecasters, as well as the use of other specialized tracking algorithms, such as the European Centre for Medium-Range Weather Forecasts (ECMWF) Tracker (Bi et al., 2023; Lam et al., 2022). To address these issues, data-driven typhoon trajectory prediction methods have been proposed (Huang et al., 2023; Rüttgers et al., 2019). Additionally, methods to generate reanalysis data for applying ECMWF Tracker (ECMWF, 2021) have also emerged (Pathak et al., 2022; Espeholt et al., 2022; Lam et al., 2022; Nguyen et al., 2023). Yet, they possess the following limitations: Weather forecasting typically uses the ECMWF ReAnalysis-v5 (ERA5) (Dee et al., 2011) data for training and inference. However, ERA5 data goes through post-correction processes, including data assimilation based on NWP forecast data. As a result, although ERA5 data shows high accuracy, it is not accessible until 3-5 days after the typhoon.

---

[*]Y.-J. Park and M.-S. Seo provided equal contributions to this work.

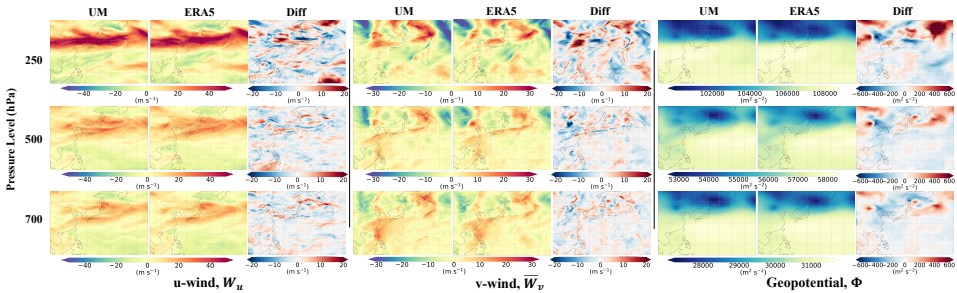

Figure 1: Visualization and comparison of the UM and ERA5 data at pressure levels of 250, 500, and 700 hPa, along with a difference map of two datasets. The u-wind and v-wind represent zonal wind and meridional wind, respectively. The UM forecast data is at a lead time of +72 hours, and the ERA5 data corresponds to that forecasted time. This analysis covers the Western North Pacific basin, with latitudes ranging from 0 to 59.75°N and longitudes from 100°E to 179.75°E.

Consequently, they are unsuitable for real-time tasks like typhoon trajectory prediction. In addition, existing data-driven real-time trajectory prediction models (Huang et al., 2022) restrict their forecasts to less than 24 hours. This considerably shorter prediction window, compared to the NWP models, makes them incapable of planning effective damage control for typhoons.

For data-driven typhoon trajectory prediction, the dataset known as *Best Track* (Knapp et al., 2010) has been a cornerstone. Documenting the intensity and central points of typhoons, its records extend back to 1950. Correspondingly, the ERA5 reanalysis data, which provides invaluable insights as an additional physics-conditioned dataset, is accessible for the same time range. In contrast, the Unified Model (UM) (Brown et al., 2012) dataset is produced in near-real time, including geopotential height, wind vectors, and other atmospheric variables using primitive equations such as the conservation of momentum, mass, energy, and water mass. However, the UM dataset inherently has several limitations.

The UM data records only include data from 2010 to the present. Furthermore, while the UM dataset adheres to physical formulas, it does not guarantee an exact representation of real-world values and can have potential errors (Brown et al., 2012). Figure 1 and Table 1 compare the UM dataset with ERA5. As illustrated, the UM data significantly differs from ERA5 in several areas. Meanwhile, as Table 1 reveals, while the UM data has only a 3-hour data acquisition delay, it exhibits larger errors.

In this study, we address these issues by introducing a Long-Term Typhoon Trajectory Prediction method (LT3P). LT3P primarily consists of two components: a physics-conditioned encoder and a trajectory predictor. The first component, the physics-conditioned encoder, encodes influential variables representing the typhoon's trajectory and intensity. In this initial phase, we harness three atmospheric variables—geopotential height, zonal, and meridional wind—across three pressure levels (700, 500, and 250 hPa) sourced from the ERA5 dataset. After this, all network parameters are frozen. The next step exclusively focuses on the bias corrector from UM data to ERA5. This correction process is fine-tuned on UM's geopotential height and wind vector data from pressure levels 250, 500, and 700 — available since 2010 — combined with the Best Track dataset. The second component, the trajectory predictor, accepts the typhoon's central coordinates as input and forecasts its trajectories for the future 72 hours. Features from the physics-conditioned encoder are cross-attended within this trajectory predictor, facilitating precise physics-conditioned typhoon trajectory predictions.

Three primary contributions can be summarized:

- We propose, for the first time, a real-time +72 hours typhoon trajectory prediction model without reanalysis data.

- We provide the preprocessed dataset *PHYSICS TRACK*, and training, evaluation, and pretrained weights of LT3P.

- In the +72 hours forecast, we achieved state-of-the-art results, outperforming the NWP-based typhoon trajectory forecasting models by significant margins.

Table 1: Comparison of ERA5 and NWP datasets: Highlighting the trade-off between real-time data availability and forecast accuracy. While ERA5 boasts high accuracy, it lacks immediacy. In contrast, NWP provides real-time data but may contain forecasting errors.

| | Type | Data Availability | Real-time | Limitations | Forecast Accuracy |
|---|---|---|---|---|---|
| ERA5 | Reanalysis Dataset | 1950 - Present | No | Time delay for data availability | $\approx 95\%$ |
| UM | NWP Model | Approx. 2010 - Present | Yes | Potential for model errors in forecasting | Low |
| IFS | NWP Model | 1974 - Present | Yes | Potential for model errors in forecasting | High |
| GFS | NWP Model | 2015 - Present | Yes | Potential for model errors in forecasting | Middle |

## 2 RELATED WORK

Typhoon trajectory prediction and human trajectory prediction are closely related in that they focus on modeling future trajectories. Unlike the trajectories of general robots and vehicles (Jiang et al., 2023), typhoon trajectories have high uncertainty because the direction and speed of movement are not deterministic (Golchoubian et al., 2023). These characteristics are similar to pedestrian trajectory prediction (Bae et al., 2022b) in that the direction and speed of pedestrian movement are not determined and its uncertainty is high.

However, while human predictions focus on capturing social interactions between agents, typhoon predictions consider overall atmospheric and geoscientific variables. In this section, we explore a typhoon trajectory prediction that takes into account the benefits of two distinct fields.

### 2.1 TRAJECTORY PREDICTION

Trajectory prediction has primarily evolved from a focus on human trajectory prediction (Alahi et al., 2016; Gupta et al., 2018; Huang et al., 2019; Mangalam et al., 2020; Salzmann et al., 2020; Gu et al., 2022), as it has been extensively applied to human-robot interaction systems (Mavrogiannis et al., 2023).

SocialGAN (Gupta et al., 2018) employs a Generative Adversarial Network (GAN) framework for predicting realistic future paths. By leveraging latent space vectors, it is able to generate various outputs that capture the intricate nuances of socially acceptable and multimodal human behaviors. PECNet (Mangalam et al., 2020) offers a distinct conditional VAE-based approach by transforming the latent spaces. It aims to strike a balance between the fidelity and diversity of predicted samples through its truncation trick in the latent space. Trajectron++ (Salzmann et al., 2020) provides a holistic methodology, predicting the probability of a latent distribution during inference. This robustness arises from its ability to adapt distribution parameters based on external input, such as prior trajectory data. Graph Neural Network (GNN)-based approaches (Huang et al., 2019; Mohamed et al., 2020; Shi et al., 2021; Bae et al., 2022a) have attracted interest in modeling and predicting trajectories in scenarios with dense populations, such as pedestrian movements. Recently, MID (Gu et al., 2022) proposes a high-performing trajectory prediction method. This method enhances the conventional trajectory prediction structure—which previously consisted only of long short-term memory (LSTM), multi-layer perceptron (MLP), and GNN—by integrating Transformer and diffusion processes.

However, when human trajectory prediction methods are applied to typhoon trajectory prediction, the results typically show poor performance (Huang et al., 2022; 2023). Unlike pedestrian scenes with multiple interacting entities, typhoons usually appear singly or in pairs at most. This difference implies that the detailed social modeling crucial for pedestrian predictions might be less important in the case of typhoons. While GNN techniques excel in complex interactive settings, their efficacy is reduced by the inherent nature of typhoons.

### 2.2 TYPHOON TRAJECTORY PREDICTION

Forecasting the trajectory of typhoons is still a complex task, affected by many variables including weather conditions, sea surface temperatures, and topography (Emanuel, 2007). As studies have progressed, forecasting methods have improved, moving from basic statistical models to more advanced contemporary numerical approaches (Wang et al., 2015; Chen & Zhang, 2019). Particularly, NWP systems, which leverage an abundance of meteorological data, have emerged as fundamental

tools for typhoon trajectory prediction. However, the interpretation of NWP model outputs often requires the expertise of meteorologists or relies on complex typhoon trajectory prediction algorithms.

On the other hand, data-driven approaches (Pathak et al., 2022; Espeholt et al., 2022; Lam et al., 2022; Nguyen et al., 2023) demonstrate notable performance in the fields of climate science, suggesting their potential applicability in forecasting typhoon trajectories. Initial attempts integrate Recurrent Neural Networks (RNNs) with specific data types but often lead to unsatisfactory results (Alemany et al., 2019). Leveraging a broader range of meteorological data has enhanced prediction accuracy, even though this introduces challenges in data processing and encoding (Giffard-Roisin et al., 2020). Notably, the recent leading approaches in typhoon trajectory prediction emphasize multi-trajectory forecasts, enabled through GANs (Huang et al., 2022; Dendorfer et al., 2021)

Despite these efforts, data-driven models still exhibit limitations, particularly with respect to their long-term forecast performance (Chen et al., 2020; Huang et al., 2023) and real-time. In this context, our work aims to pioneer long-term predictions by integrating physical values from UM data, which are outputs from a real-time-capable NWP model, with the power of data-driven methods.

## 3    METHOD

The LT3P model takes the typhoon center coordinates $C_o := \{c\}_{i=1}^{t_o} = \{x_{\text{lon}}, y_{\text{lat}}\}_{i=1}^{t_o}$, where $t_o$ represent the input time sequences, while $x_{\text{lon}}$ and $y_{\text{lat}}$ denote the longitude and latitude. Here, $i$ represents the index in the time sequence. In LT3P, the inputs are the geopotential height and wind vector from the UM, represented as $\Phi_P = \{z_p\}_{p \in P, i=t_o+1}^{t_o+t_f}$ and $\vec{W}_P = \{w_p\}_{p \in P, i=t_o+1}^{t_o+t_f}$, respectively. Here, $P$ refers to the pressure levels and we use it as $P = \{250, 500, 700\}$hPa. $t_f$ represent the output time sequences. LT3P then outputs the future typhoon center coordinates $C_f := \{u\}_{i=t_o+1}^{t_o+t_f} = \{x_{lon}, y_{lat}\}_{i=t_o+1}^{t_o+t_f}$.

Hence, LT3P comprises a physics-conditioned model which extracts the representation from $\Phi$ and $\vec{W}$, and a trajectory predictor that forecasts the typhoon trajectory. LT3P is optimized through a dual-branch training strategy. During a pre-trained phase, the physics-conditioned model is trained using ERA5 as in (Nguyen et al., 2023). Subsequently, in a bias correction phase, the trajectory predictor is trained to produce outputs akin to ERA5, even when UM data is used as the input.

### 3.1    PRELIMINARIES

The NWP forecasting system primarily hinges on a core prediction formula for the geopotential height $\Phi$ and the wind vector $\vec{W}$ based on conservation laws:

$$\frac{\partial}{\partial t} \begin{bmatrix} \Phi, & \vec{W}, & \rho, & T, & q \end{bmatrix} = \mathcal{F}\left(\vec{W}, \Phi, \rho, T, q\right), \tag{1}$$

where $\mathcal{F}$ refers to interactions among variables following the conservation laws. The variables $\rho, T$, and $q$ denote air density, temperature, and specific humidity, respectively.

Conservation equations underpin the NWP's forecasting. For instance, a momentum conservation with the spatial gradient $\nabla$ is expressed by:

$$\frac{\partial \vec{W}}{\partial t} = -\vec{W} \cdot \nabla \vec{W} \frac{\nabla P}{\rho} + \vec{g}, \tag{2}$$

and a mass conservation is below:

$$\frac{\partial \rho}{\partial t} + \nabla \cdot (\rho \vec{W}) = 0. \tag{3}$$

Other conserved quantities include energy, related to geopotential height $\Phi$ and temperature $T$, and water vapor content or specific humidity $q$.

**Note on Accuracy and Time $t$.** In the NWP system, a time $t$ indicates progression. For longer predictions up to +72 hours, their accuracy depends on various factors: initial conditions, physics parameterization schemes, spatial resolution, and so on. While the conservation law captures large-scale atmospheric dynamics effectively, it might not work on small-scale details due to local conditions like microclimates, terrain variations, or transient phenomena.

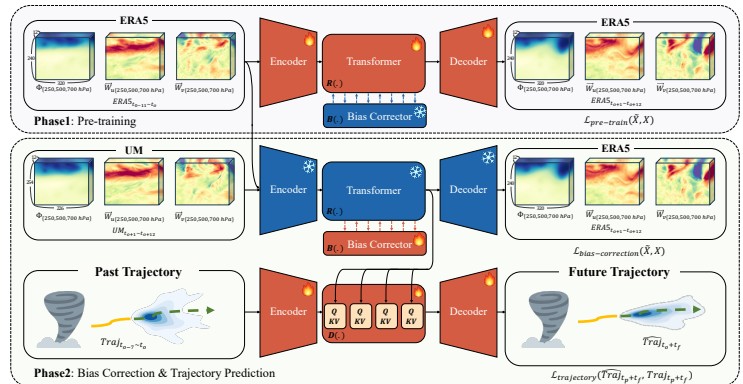

Figure 2: Overview of LT3P. In phase 1, the physics-conditioned model $R(\cdot)$ is trained for the weather forecasting task to encode information from $\Phi$, $\vec{W}_u$, and $\vec{W}_v$. Subsequently, in phase 2, using UM as the input, the typhoon trajectory prediction model $D(\cdot)$ is trained in conjunction with the bias-corrector $B(\cdot)$ to make accurate predictions.

The Earth's atmosphere is a chaotic system (Lorenz, 1963). Despite relying on conservation laws, NWP models are inevitably prone to errors due to the inherent imperfections in our understanding of physical processes and the true state of the atmosphere. These discrepancies underscore the importance of data-driven approaches, in spite of the potential error in the UM data.

## 3.2 PRE-TRAINING ON ERA5 DATASET

The ERA5 dataset, accumulated at 6-hour intervals from 1950 to the present, requires a significant amount of time to train a model from scratch. To address this challenge, we have adopted a strategy of constructing a foundational model for typhoon trajectory prediction.

The physics-conditioned model of LT3P is trained for a weather forecasting task, similar to (Nguyen et al., 2023), to effectively learn the representations of the highly-related variables for typhoon trajectory prediction in (Giffard-Roisin et al., 2020): geopotential height($\Phi$), zonal wind($\vec{W}_u$), and meridional wind($\vec{W}_v$).

As shown in Figure 2, the physics-conditioned model consists of an encoder-transformer-decoder architecture. The encoder and decoder are built using 3D convolutions for spatio-temporal modeling, and the transformer is also constructed as a 3D transformer.

The objective of the physics-conditioned model is as follows:

$$L_{pre\_train} = \frac{1}{T \times V \times H \times W} \sum_{i=1}^{T} \sum_{j=1}^{V} \sum_{k=1}^{H} \sum_{l=1}^{W} (\tilde{X}_{t_f}^{i,j,k,l} - X_{t_f}^{i,j,k,l})^2, \qquad (4)$$

where $T$ is the final lead time, $V$ represents 9 variables, with $\Phi$, $\vec{W}_u$, and $\vec{W}_v$ for each of the 3 pressure levels. $H$ and $W$ denote height and width, respectively. $\tilde{X}$ is the forecasting value for ERA5 from time $t$ to $t + t_f$ in the physics-conditioned model, given $V$ as input from the period $t - t_f$ to $t$. $X$ is its ground truth (GT).

## 3.3 BIAS CORRECTION & TYPHOON TRAJECTORY PREDICTION

**Bias correction.** Since ERA5 is reanalysis data, this limits the applicability on real-time tasks, which makes us use the UM data. Hence, the typhoon trajectory prediction model, LT3P, accepts two primary inputs: the representation encoded by the $R(\cdot)$ —derived from the physics formula presented in Section. 3.1 using $\Phi$ and $\vec{W}$—and the typhoon's center coordinates $C_o$. The model then produces the output $C_f$.

However, since our $R(\cdot)$ was trained with the ERA5 data, it may struggle with extracting meaningful representations from the UM data due to their domain gap. To solve this problem, LT3P uses the

Table 2: Ensemble typhoon trajectory prediction results (FDE): Note that while the original MGTCF study only includes 24-hour predictions, for comparison purposes, we modified it to input 48 hours at 6-hour intervals and predict +72 hours at 6-hour intervals. **Bold** indicates the best performance, and underline indicates the second-best performance. Note that **Dataset** refers to the dataset used for inference.

| Method | Real-time | Dataset | | | Distance (km) | | | | | | | | | | | |
|---|---|---|---|---|---|---|---|---|---|---|---|---|---|---|---|---|
| | | BST | ERA5 | UM | 6 h | 12 h | 18 h | 24 h | 30 h | 36 h | 42 h | 48 h | 54 h | 60 h | 66 h | 72 h |
| SocialGAN (Gupta et al., 2018)-Ens | ✓ | ✓ | - | - | 33.16 | 63.25 | 93.33 | 131.31 | 176.71 | 226.24 | 288.78 | 353.79 | 431.03 | 519.93 | 611.67 | 716.90 |
| STGAT (Huang et al., 2019)-Ens | ✓ | ✓ | - | - | 56.68 | 100.51 | 160.08 | 209.83 | 269.81 | 325.44 | 377.86 | 435.69 | 492.44 | 550.37 | 613.97 | 682.16 |
| PECNet (Mangalam et al., 2020)-Ens | ✓ | ✓ | - | - | 41.61 | 56.74 | 82.26 | 112.28 | 147.96 | 185.92 | 232.79 | 279.29 | 330.84 | 388.25 | 445.61 | 518.46 |
| MID (Gu et al., 2022)-Ens | ✓ | ✓ | - | - | 109.31 | 173.35 | 230.45 | 289.62 | 359.10 | 422.11 | 490.93 | 564.23 | 638.04 | 712.97 | 798.63 | 881.34 |
| MMSTN (Huang et al., 2022)-Ens | ✓ | ✓ | - | - | 49.72 | 109.23 | 173.64 | 246.26 | 334.62 | 430.45 | 552.15 | 667.34 | 819.22 | 969.13 | 1126.26 | 1300.59 |
| MGTCF (Huang et al., 2023)-Ens | ✗ | ✓ | ✓ | - | 46.12 | 81.68 | 116.81 | 155.72 | 201.26 | 249.42 | 302.19 | 361.14 | 427.35 | 499.85 | 576.80 | 655.48 |
| JTWC (Chen et al., 2023) | ✓ | - | - | - | - | - | - | 87.09 | - | - | - | 146.45 | - | - | - | 217.05 |
| JMA-GEPS (Chen et al., 2023) | ✓ | - | - | - | - | 62.83 | - | 98.10 | - | 138.26 | - | 180.04 | - | 222.75 | - | 259.07 |
| ECMWF-EPS (Chen et al., 2023) | ✓ | - | - | - | - | 57.75 | - | 78.87 | - | 108.28 | - | 138.97 | - | 176.02 | - | 210.71 |
| NCEP-GEFS (Chen et al., 2023) | ✓ | - | - | - | - | 52.34 | - | 76.17 | - | 105.55 | - | 142.26 | - | 185.36 | - | 225.59 |
| UKMO-EPS (Chen et al., 2023) | ✓ | - | - | - | - | 62.74 | - | 90.90 | - | 122.90 | - | 160.41 | - | 200.89 | - | 245.28 |
| LT3P (UM Only)-Ens | ✓ | ✓ | - | ✓ | 42.20 | 58.83 | 88.39 | 114.84 | 147.39 | 189.03 | 220.41 | 255.57 | 320.47 | 334.88 | 370.64 | 390.92 |
| LT3P (Bias-corrected UM)-Ens | ✓ | ✓ | - | ✓ | 6.30 | 19.45 | 22.83 | 39.81 | 50.29 | 60.76 | 65.38 | 70.01 | 83.98 | 100.88 | 110.36 | 143.03 |
| LT3P (ERA5 Only)-Ens | ✗ | ✓ | ✓ | - | **5.29** | **6.80** | **12.34** | **19.90** | **24.09** | **30.42** | **35.92** | **39.29** | **44.83** | **50.30** | **60.11** | **70.94** |

* Note that best track data from MMSTN and MGTCF could not be obtained because they use data from specific organizations. Therefore, the experimental results of each of the two studies were evaluated only in 2019.

bias-corrector $B(\cdot)$ to adapt the variables generated through UM to the variables of ERA5. The $B(\cdot)$ has the following objective functions:

$$L_{Bias\_correction} = \frac{1}{T \times V \times H \times W} \sum_{i=1}^{T} \sum_{j=1}^{V} \sum_{k=1}^{H} \sum_{l=1}^{W} (\hat{X}_{t_f}^{i,j,k,l} - X_{t_f}^{i,j,k,l})^2, \tag{5}$$

where $\hat{X}$ is the ERA5 prediction which is bias-corrected version of the UM dataset.

**Typhoon Trajectory Prediction.** In the second phase, LT3P takes ERA5 data from 1950 to 2010 and UM data from 2010 to 2018 as inputs in order to perform typhoon trajectory prediction. The main model of LT3P, which predicts the center coordinates of the typhoon, generates the final trajectory by applying a cross-attention between the 3D features from the $R(\cdot)$'s transformer and the 1D features of the typhoon coordinates from the LT3P model's encoder. The cross attention enables LT3P to take fully advantage of both data-driven and physics-based prediction. It also allows us to fuse these two features regardless of the shape or length of the features. The cross attention is defined as:

$$\text{Attention}(Q, K, V) = \text{softmax}\left(\frac{QK^T}{\sqrt{d}}\right) \cdot V,$$

where $Q = W_Q^{(i)} \cdot \phi_i(R(\Phi, \vec{W}))$, $K = W_K^{(i)} \cdot \tau_\theta(C_p)$, and $V = W_V^{(i)} \cdot \tau_\theta(C_p)$. The $\theta$ represents the set of parameters associated with the $D(\cdot)$. Here, $\phi_i(z_t) \in \mathbb{R}^{N \times d_i}$ denotes a (flattened) intermediate representation of the transformer implementing $\theta$. $W_V^{(i)} \in \mathbb{R}^{d \times d_i}$, $W_Q^{(i)} \in \mathbb{R}^{d \times d_\tau}$, and $W_K^{(i)} \in \mathbb{R}^{d \times d_\tau}$ are learnable projection matrices.

Therefore, in LT3P, the final objective function is defined as follows:

$$L_{total} = L_{Bias\_correction} + L_{trajectory}. \tag{6}$$

Note that various trajectory prediction baselines such as GAN, CVAE, and diffusion-based models can be applied, whose effectiveness will be validated in section 4.3.

## 4 EVALUATIONS

In this section, we first describe the dataset and the pre-processing methods used as well as an implementation detail of our LT3P. We then show quantitative and qualitative evaluations and provide an extensive ablation study to check the effectiveness of each component in LT3P.

### 4.1 EXPERIMENTAL SETTINGS

***Physics Track* dataset.** We collect the center coordinates of typhoons at 6-hour intervals (00, 06, 12, 18) from 1950 to 2021. We use data corresponding to the Western North Pacific basin, with latitudes ranging from 0 to 59.75°N and longitudes from 100°E to 179.75°E, as typhoons overwhelmingly

Table 3: Results of Stochastic Typhoon Trajectory Predictions (FDE): This table presents results generated from 20 samples, consistent with the conventional approach in NWP-based GEFS (Zhou et al., 2022), which employs 20 ensemble members for the final prediction.

| Method | Distance (km) | | | | | | | | | | | |
|---|---|---|---|---|---|---|---|---|---|---|---|---|
| | 6 h | 12 h | 18 h | 24 h | 30 h | 36 h | 42 h | 48 h | 54 h | 60 h | 66 h | 72 h |
| SocialGAN (Gupta et al., 2018) | 17.59 | 30.30 | 40.97 | 50.30 | 64.40 | 71.63 | 88.00 | 97.81 | 110.49 | 126.92 | 144.66 | 164.99 |
| STGAT (Huang et al., 2019) | 32.81 | 55.24 | 86.86 | 110.93 | 147.86 | 180.03 | 204.79 | 233.53 | 259.93 | 277.09 | 298.62 | 322.54 |
| PECNet (Mangalam et al., 2020) | 28.40 | 34.34 | 45.88 | 57.36 | 69.44 | 81.17 | 97.84 | 112.34 | 130.22 | 150.18 | 169.22 | 188.74 |
| MID (Gu et al., 2022) | 24.54 | 42.66 | 56.52 | 73.41 | 96.04 | 116.86 | 136.44 | 160.55 | 184.09 | 205.78 | 230.09 | 256.26 |
| MMSTN (Huang et al., 2022) | 19.46 | 42.44 | 67.38 | 93.59 | 128.97 | 162.90 | 193.29 | 232.58 | 269.17 | 309.55 | 348.10 | 384.57 |
| MGTCF (Huang et al., 2023) | 20.84 | 37.59 | 52.45 | 67.29 | 83.09 | 104.13 | 128.90 | 156.98 | 186.62 | 218.18 | 249.22 | 280.90 |
| LT3P (Bias-corrected UM) | 1.97 | 2.31 | 3.32 | 8.80 | **17.89** | 25.53 | 30.57 | **36.39** | 41.12 | **47.38** | **51.87** | **65.24** |
| LT3P (ERA5 Only) | **0.66** | **1.12** | **2.26** | **2.34** | 21.93 | 26.36 | 26.99 | 41.39 | 40.24 | 49.63 | 52.97 | 65.95 |

\* Note that best track data from MMSTN and MGTCF could not be obtained because they use data from specific organizations. Therefore, the experimental results of each of the two studies were evaluated only in 2019.

occur within this region. (Stull, 2011) Any typhoon that dissipated within 48 hours is excluded from the dataset. The training data uses years 1950-2018 and comprises 1,334 typhoons, while the test dataset covers years 2019-2021 and includes 90 typhoons. Note that for hyperparameter tuning, we train on data from 1950 to 2016 and set the 2017 to 2018 data as the validation set. Subsequently, we evaluate the final performance of model using the entire dataset from 1950 to 2018.

In sync with the duration of the typhoon, we utilize variables in the ERA5 dataset which are closely associated with typhoons, notably the geopotential height and wind vectors. Data for the pressure levels at 250, 500, and 700 hPa are collected. For training purposes, the ERA5 dataset is employed from 1950 to 2018, and the data spanning 2019, 2020, and 2021 is set aside. Regarding the real-time data from the UM, we compile data from the years 2010 to 2021, ensuring the inclusion of the same variables and pressure levels as found in the ERA5 dataset.

*Note:* All variables in ERA5 and UM are normalized using the formula $(\text{variable} - \text{mean})/\text{std}$ because no specific minimum-maximum range is predetermined for each variable.

For both the ERA5 and UM datasets, information is extracted to match the conditions of the best track. Therefore, the input dimension for ERA5 and UM is set to $(B \times T \times V \times H \times W) \rightarrow (B \times 12 \times 9 \times 240 \times 320)$. Note that the resolution has been adjusted using bi-linear interpolation.

**Evaluation metrics.** We compare our LT3P with global meteorological agencies, including the Joint Typhoon Warning Center (JTWC), Japan Meteorological Agency (JMA), European Centre for Medium-Range Weather Forecasts (ECMWF), National Centers for Environmental Prediction (NCEP), and the United Kingdom Meteorological Office (UKMO). Additionally, the Global Ensemble Forecast System (GEFS) refers to the results of 20 separate generated forecasts (Zhou et al., 2022) and then combines them into an ensemble average forecast, which is to average the sampled forecasting results. Therefore, the data-driven models are evaluated using the ensemble average method to benchmark their performance compared to the ensemble average NWP models. Performance is measured at 6-hour intervals, consistent with existing meteorological agencies.

We use (1) Average Displacement Error (ADE) - average Euclidean distance between a prediction and ground-truth trajectory; (2) Final Displacement Error (FDE) - Euclidean distance between a prediction and ground-truth final destination.

**Implementation details.** The physics-conditioned model $R(\cdot)$ is comprised of an encoder, translator and decoder. The encoder and decoder are composed of (`Conv, LayerNorm, SiLU`) (Elfwing et al., 2018)) and (`Conv, LayerNorm, SiLU, PixelShuffle`), respectively. In the experiments, the encoder and decoder consist of four blocks each. The translator is selected as the transformer block and has a total of three blocks. The trajectory predictor $D(\cdot)$ follows GAN (Gupta et al., 2018), CVAE (Mangalam et al., 2020) and diffusion (Gu et al., 2022), respectively, and the experimental results in Table 2 and Table 3 are obtained from the diffusion-based predictor.

For training, we use a machine with 8 Nvidia-Quadro RTX 8000 GPUs. The batch size is set to 128, and we use Adam optimizer with a learning rate of 0.001. Additionally, an Exponential Moving Average with a momentum of 0.999 is applied, and the training is conducted for 2,000 epochs. Note that we did not employ data augmentation, as the geometric and intensity values in typhoon forecasting are extremely sensitive. Applying data augmentation might compromise the integrity of physical rules. For fair comparison, the data-driven baseline models such as SocialGAN (Gupta et al., 2018), STGAT (Huang et al., 2019), PECNet (Mangalam et al., 2020), MID (Gu et al., 2022), and MMSTN (Huang et al., 2022) are trained and evaluated on the best track dataset, while MGTCF (Huang et al., 2023) utilizes both the best track and ERA5 datasets.

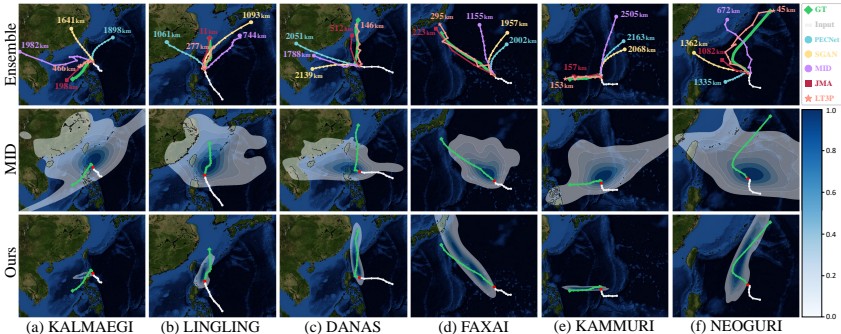

Figure 3: Qualitative analysis between LT3P and other trajectory prediction baselines. Note that the typhoon trajectory provided by JMA exists at 24-hour intervals, so the three coordinates have been linearly connected. Additionally, the error (in km) of the final coordinate is also indicated. The probability map is visualized using kernel density estimation (KDE).

## 4.2 EXPERIMENTAL RESULTS

**Quantitative results.** In Table 2, we compare the performance of LT3P with the state-of-the-art human trajectory prediction methods as well as data-driven typhoon prediction methods. The table also includes forecast results from several renowned meteorological agencies, including JTWC, JMA, ECMWF, NCEP, and UKMO, which are based on the conservation laws in Section. 3.2. Our observations indicate that LT3P (ERA5 Only) exhibits the most promising results, with LT3P (Bias-corrected UM) trailing closely behind. It is important to highlight that MID, which has achieved the best results among data-driven human trajectory predictors, performs poorly in ensemble forecasting. This discrepancy arises from the inherent goal of human trajectory prediction models. They have developed to capture social relations between agents, and its probabilistic distribution attempts to cover a wide and diverse distribution to represent all feasible socially-acceptable paths and destination.

Furthermore, it is noteworthy that our model consistently outperforms the numerical prediction models of the established operational forecast centers. These experimental results demonstrate the potential of real-time NWP data to serve as an alternative to reanalysis data in the field of real-time typhoon trajectory prediction. Note that studies in (Bi et al., 2023; Lam et al., 2022), which predict reanalysis data and then use the ECMWF-tracker for typhoon trajectory prediction, are excluded in our comparisons because code and weights are unavailable, and there are no access to the ECMWF-tracker (ECMWF, 2021).

Table 3 shows the performance results from our stochastic trajectory prediction and the existing approaches to data-driven trajectory prediction. We report the results have the lowest error among 20 generated trajectories. As revealed in Table 3, LT3P stands out as the state-of-the-art model for stochastic trajectory prediction. We also observe significant performance improvements in other data-driven trajectory prediction models. These encouraging results can be attributed to the inherent strength of existing data-driven methods, which are designed to generate a diverse and wide range of trajectory predictions.

**Qualitative Results** Figure 3 presents the qualitative results for ensemble and stochastic predictions. In Figure 3-(c), only the JMA model and our LT3P predict the change in trajectory of typhoons like DANAS well. These results highlight the importance of physics values in typhoon trajectory prediction. In Figure 3-(f), LT3P superiority is also confirmed as well. In addition, MID, using only coordinates, outperforms the physics-based JMA. Such instances show the promising potential of data-driven models. Additionally, in the case of stochastic results, LT3P does not aim for a wide and diverse range of predictions, but focuses on generating forecasts that closely approximate the actual typhoon trajectory. These qualitative results are supporting evidence for LT3P's strong performance in ensemble forecasting, as reported in Table 2.

Figure 4 presents the differences in zonal wind field between UM 72-hour forecast and ERA5 before and after the bias correction of UM. The figure shows that the UM exhibits a significantly smaller bias

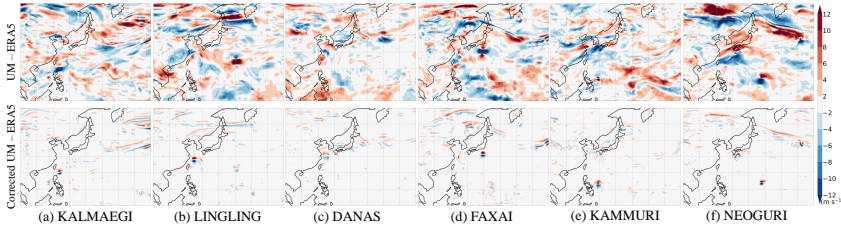

Figure 4: Zonal wind bias of the UM before and after bias correction with respect to ERA5.

Table 4: Ablation study on the efficiency of each component of LT3P.

| Component | | | | Metric |
|---|---|---|---|---|
| UM | Joint Training | Pre-Training | Bias-correction | ADE/FDE (km) |
| ✓ | | | | 219.46 / 390.92 |
| ✓ | ✓ | | | 80.39 / 190.75 |
| ✓ | ✓ | ✓ | | 85.62 / 198.11 |
| ✓ | ✓ | ✓ | ✓ | **65.25 / 143.03** |

Table 5: Ablation study on backbone models.

| Backbone | Metric |
|---|---|
| | ADE/FDE (km) |
| SocialGAN (Gupta et al., 2018) | 69.99 / 155.23 |
| PECNet (Mangalam et al., 2020) | 69.81 / 152.07 |
| MID (Gu et al., 2022) | **65.25 / 143.03** |

after the bias-correction phase than the original UM field. These show that the atmospheric conditions across the entire region, including the trajectories of the typhoons, have been well corrected in LT3P.

## 4.3 ABLATION STUDY

To assess the efficacy of each component within the LT3P framework, a systematic analysis is conducted: a baseline trained solely on the UM, a joint training scheme trained on ERA5 from 1950-2009 and the UM from 2010-2018, an extended method incorporating phase 1 pre-training and, the full LT3P pipeline with the bias correction. The results are reported in Table 4. The empirical findings in Table 4 suggest that all components, barring the UM Only training, yield good results. In particular, the LT3P configuration, which leverages all the components, achieves the best performance.

Table 5 presents the experimental results when applying well-known trajectory prediction models such as GAN, CVAE, and diffusion-based methods to the LT3P framework. In Table 5, the diffusion-based method achieves the better performance than the others. Moreover, all the methods, including GAN, CVAE, and diffusion, exhibit significant performance improvements when incorporating UM data compared to using only coordinate data (refer to the prediction result on 72 hours in Table 2). These findings indicate that the performance gains can be attributed to the LT3P framework itself rather than to the backbone models.

## 5 CONCLUSION

We propose the Long-Term Typhoon Trajectory Prediction (LT3P) for real-time typhoon trajectory prediction without reanalysis data. To the best of our knowledge, LT3P is the first data-driven typhoon trajectory prediction that employs a real-time NWP dataset. Unlike methods that generate reanalysis data, LT3P predicts the typhoon's central coordinates without any need for an additional forecaster and algorithms. In addition, our LT3P does not require weather forecasting experts, making it easily accessible for various institutions with limited meteorological infrastructure. Using extensive evaluations, we confirm that our LT3P achieved state-of-the-art performance in typhoon trajectory prediction. However, our model has only been applied to typhoons and has not been applied to other regions of tropical cyclones. In future work, we plan to apply it to all kinds of tropical cyclones. Lastly, we hope to contribute to the field of climate AI by releasing our *PHYSICS TRACK* dataset, training, test codes of LT3P, and pretrained weights to the public.

## ACKNOWLEDGMENTS

This work is in part supported by the Institute of Information & communications Technology Planning & Evaluation (IITP) grant funded by the Korea government (MSIT) (No.2019-0-01842, Artificial Intelligence Graduate School Program (GIST) and No.2021-0-02068, Artificial Intelligence Innovation Hub), and GIST-MIT Research Collaboration grant funded by the GIST in 2024.

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

# A   APPENDIX

## A.1   TRAJECTORY PREDICTOR IMPLEMENTATION DETAILS.

**SocialGAN** Gupta et al. (2018)   SocialGAN comprises sequential LSTM and MLP-based encoders, decoders, discriminators, and a pooling module.

The hidden dimension for both the encoder and decoder, as well as the bottleneck dimension, is set to 32, while the discriminator's hidden dimension is set to 64.

Furthermore, the encoder and decoder of SocialGAN are configured with a single LSTM layer.

The LT3P, utilizing the GAN-based predictor SocialGAN, adheres to the $\mathcal{L}_{trajectory}$ equation as follows:

$$\mathcal{L}_{trajectory} = \min_{k} \|Traj_{t_p+t_f} - \hat{Traj}^{(k)}_{t_p+t_f}\|_2, \tag{7}$$

where $k$ is set to 10. To implement SocialGAN, we used the official repository[1].

**PECNet** Mangalam et al. (2020)   To construct a typhoon trajectory predictor based on a conditional variational autoencoder (CVAE), we have chosen PECNet.

PECNet consists of an encoder $E_{\mathrm{past}}$ for encoding past trajectory coordinates, an encoder $E_{\mathrm{way}}$ used for predicting waypoints, a latent encoder $E_{\mathrm{latent}}$, a decoder $D_{\mathrm{latent}}$, and $P_{\mathrm{predict}}$ for predicting future trajectories.

The network architecture of PECNet is as follows: $E_{\mathrm{way}}$ has channel dimensions 2-8-16-16, $E_{\mathrm{past}}$ has dimensions 16-512-256-16, $E_{\mathrm{latent}}$ has dimensions 32-8-50-32, $D_{\mathrm{latent}}$ has dimensions 32-1024-512-1024-2, and $P_{\mathrm{predict}}$ has dimensions 32-1024-512-256-22.

All the sub-networks are multi-layer perceptrons with ReLU non-linearity.

The LT3P's loss function $\mathcal{L}_{trajectory}$ based on PECNet is defined by the equation below:

$$\mathcal{L}_{trajectory} = \lambda_1 D_{KL}(\mathcal{N}(\mu,\sigma)\|\mathcal{N}(0,\mathbf{I})) + \lambda_2 \left\|\mathcal{G}_c - \hat{\mathcal{G}}_c\right\|_2^2 + \left\|\mathcal{T}_f - \hat{\mathcal{T}}_f\right\|_2^2 \tag{8}$$

where $\mathcal{G}$ is the endpoint, and $\mathcal{T}$ represents the trajectories.

The first term is the KL divergence term used for training the variational autoencoder, the second term average endpoint loss (AEL) trains $E_{\mathrm{end}}$, $E_{\mathrm{past}}$, $E_{\mathrm{latent}}$, and $D_{\mathrm{latent}}$, and the third term average trajectory loss (ATL) trains the entire module together.

We integrated the LT3P with PECNet using the official code[2].

**MID** Gu et al. (2022)   To design a typhoon trajectory predictor based on diffusion, we select MID.

Particularly, MID possesses a transformer-based encoder and decoder.

We devise a three-layer transformer as the core network for our MID, where the transformer dimension is set to 512, and 4 attention heads are applied.

We employ one fully-connected layer to upsample the input of the model from dimension 2 to the transformer dimension and another fully-connected layer to upsample the observed trajectory feature f to the same dimension.

We utilize three fully-connected layers to progressively downsample the Transformer output sequence to the predicted trajectory, achieving dimensions of 512-256-2.

The loss function of the forward process of MID follows the equation below:

---

[1]https://github.com/agrimgupta92/sgan
[2]https://github.com/HarshayuGirase/Human-Path-Prediction

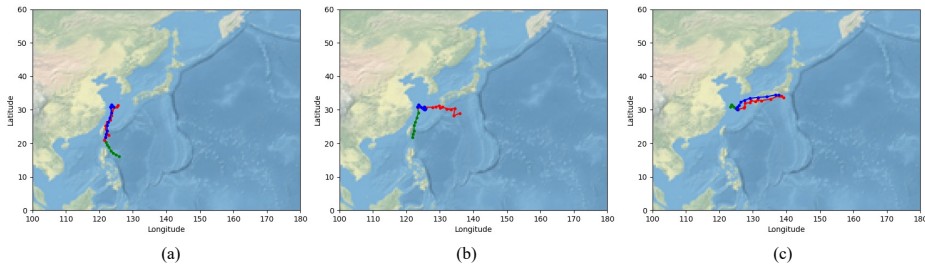

Figure 5: Result of applying LT3P to Typhoon CHANTHU, which occurred in 2021. Green dots represent the inputs, and blue dots indicate the Ground Truth (GT), and red dots are the prediction results.

$$L(\theta, \psi) = \mathbb{E}_{\epsilon, y_0, k} \left\| \epsilon - \mathcal{E}(\theta, \psi)(y_k, k, \mathcal{X}, z) \right\|, \tag{9}$$

where $\epsilon \sim N(0, I)$, $y_k = \sqrt{\bar{\alpha}_k} y_0 + \sqrt{1 - \bar{\alpha}_k}$ and the training is performed at each step $k \in \{1, 2, \ldots, K\}$. We have set $k$ to 20.

We used the official code[3] for implementation, and we also provide the code for the MID-based LT3P [4].

subsectionPhysics Track Dataset Details.

Table ?? presents the statistics of our physics track dataset. Although the total number of entries in the ERA5 dataset is 171,348, our dataset begins from various time points $t$ and extends to +72 hours as input, resulting in even more combinations.

Table 6: Statistics of the Physics track dataset. Note that UM data has +72 forecasting values at each point in time, so the total amount of data is large.

|  | Number of Samples | Time period | Data shape $(V, H, W)$ | Data size |
|---|---|---|---|---|
| UM | 525,588 | 2010~2021 | $(9 \times 254 \times 226)$ | 676G |
| ERA5 | 171,348 | 1950~2021 | $(9 \times 240 \times 320)$ | 150G |
| Best Track | 1,424 | 1950~2021 | $(T \times X \times Y)$ | 6.6M |

Figure 6 displays samples from the Best Track for the years 2019, 2020, and 2021.

As evident from the illustration, there's significant uncertainty regarding the direction of a typhoon's movement.

Typhoon durations can also vary widely, with some lasting over two weeks while others persist for as briefly as three days. This variability underscores the complexity and challenges involved in predicting typhoon trajectories.

Figure 7 is the project page for the Physics track dataset.

We will provide individual data for typhoons occurring each year, along with their corresponding ERA5 and UM data.

Not only that, but we will also make the dataset available for download in its entirety.

Furthermore, explanations for each modality will be included on the project page.

## A.2 INFERENCE IN OTHER TYPES OF TROPICAL CYCLONES

This paper focuses on the trajectory of typhoons, which is one of tropical cyclones occurring in the North Western Pacific. However, tropical cyclones also occur in various other regions.

---

[3]https://github.com/Gutianpei/MID
[4]https://github.com/iclr2024submit/LT3P

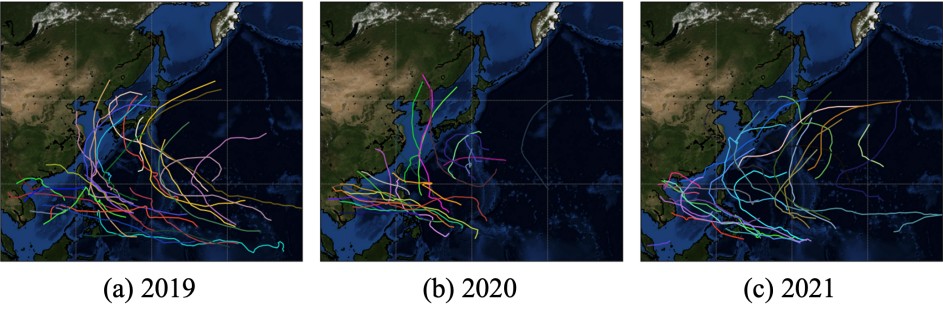

| (a) 2019 | (b) 2020 | (c) 2021 |

Figure 6: The image visualizes the typhoon trajectories from the Joint Typhoon Warning Center (JTWC)'s Best Track for the years 2019, 2020, and 2021 that we utilized.

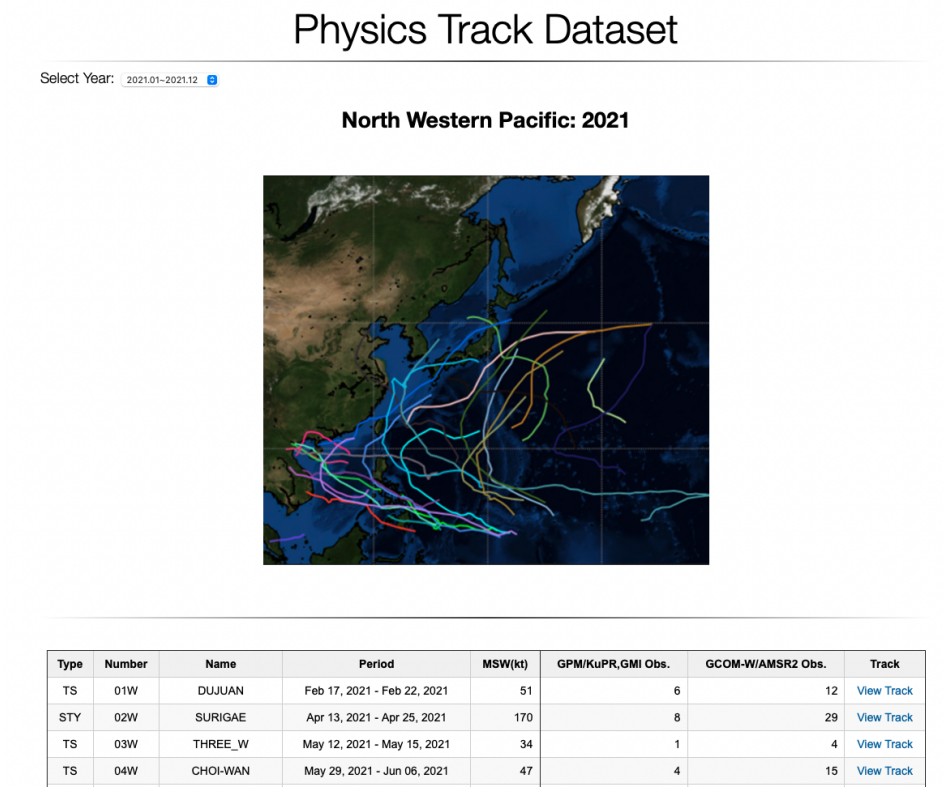

Figure 7: Physics track dataset project page overview.

To assess the potential of our LT3P model in areas outside the North Western Pacific, we conduct zero-shot validations on TAUKTAE and CHALANE, which occur in the North Indian Ocean and the South Indian Ocean, respectively, using a model trained only on the North Western Pacific.

As shown in  Figure 9, despite not being trained on the North Indian Ocean and South Indian Ocean, the model retains some level of performance.

This experimental result suggests that the performance is consistent because LT3P uses UM prediction values, which follow physical rules and thus do not exhibit large errors.

However, due to regional differences in the characteristics of tropical cyclones, the model exhibits larger errors than the average error for typhoons in the North Western Pacific.

These results indicate that additional training is required for LT3P to operate effectively in other regions.

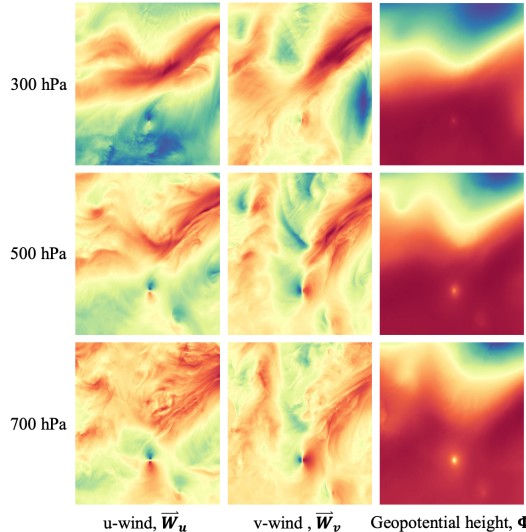

Figure 8: Visualization of all variables in the UM dataset.

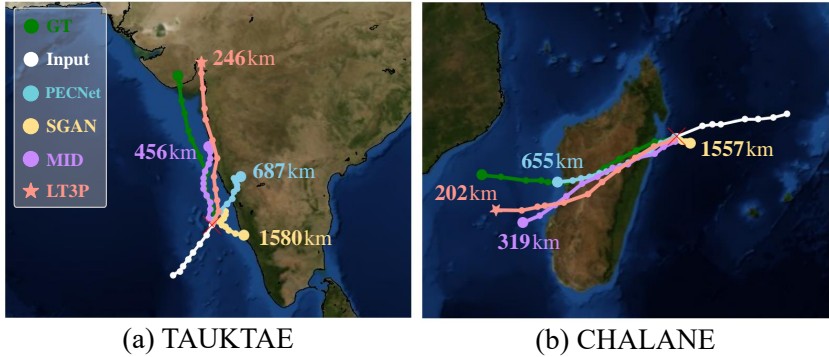

(a) TAUKTAE                    (b) CHALANE

Figure 9: Inferences on different types of tropical cyclones using LT3P and other trajectory prediction baselines.

Nevertheless, the fact that the performance is somewhat maintained suggests that LT3P has the potential to function properly in different regions.

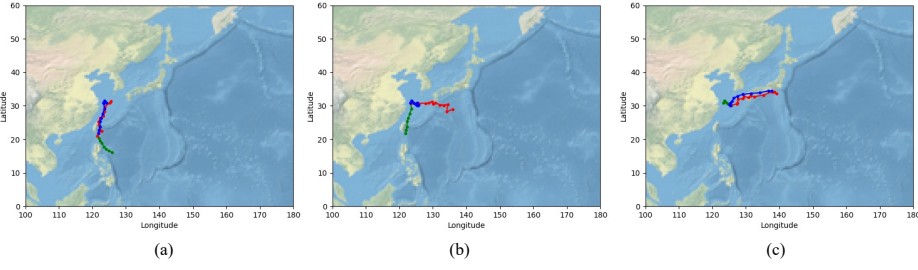

(a)                    (b)                    (c)

Figure 10: Result of applying LT3P to Typhoon CHANTHU, which occurred in 2021. Green dots represent the inputs, and blue dots indicate the Ground Truth (GT), and red dots are the prediction results.

### A.3 FAILURE CASE ANALYSIS.

Figure 10 presents the qualitative analysis results for Typhoon CHANTHU, which occurred in 2021. As shown in the figure, even though it is the same the Typhoon CHANTHU, LT3P predicted its trajectory well in the early stage (Figure 10-(a)) and in the late stage (Figure 10-(c)), but not as accurately during the middle phase, as shown in Figure 10-(b). Particularly, as seen in Figure 10-(b), we discovered that LT3P's performance degrades when the typhoon remains stationary for a long time. We speculate that this experimental result is due to the scarcity of data on stationary typhoons within the dataset.

### A.4 TEMPORAL CORRELATION COEFFICIENT

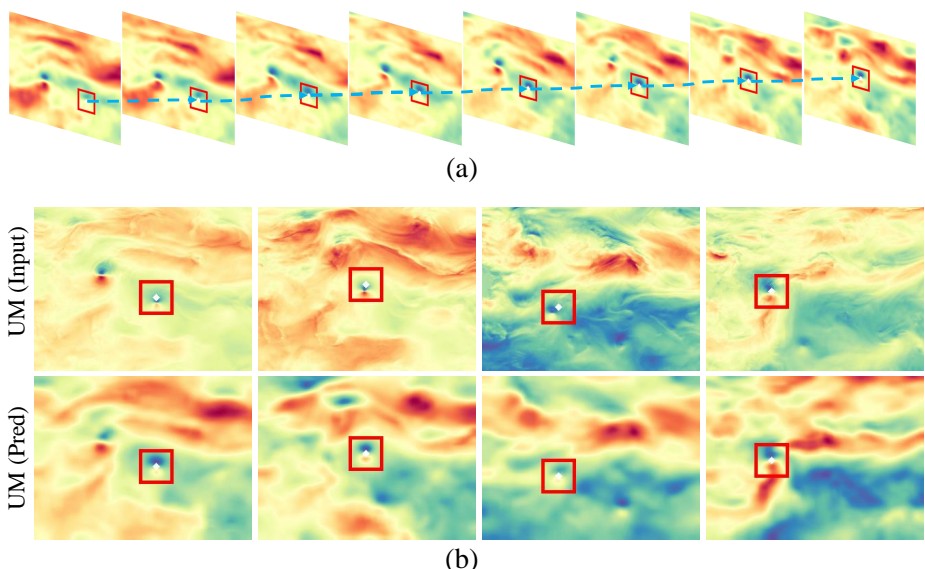

Figure 11: Qualitative analysis results for bias-corrected UM. (a) provides an overview of TCC calculation, while (b) demonstrates how the bias-corrected UM corrects the position of the center of the typhoon. The white diamond represents the center of the typhoon, and the red bounding box indicates the cropped area around the coordinate center.

To calculate the temporal correlation coefficient (TCC) scores between bias-corrected UM and ERA5 image sequences, we first compute the frame-by-frame differences:

$$\Delta \mathbf{P} = \mathbf{P}_{i+1} - \mathbf{P}_i \tag{10}$$

$$\Delta \mathbf{G} = \mathbf{G}_{i+1} - \mathbf{G}_i \tag{11}$$

where $\mathbf{P}$ and $\mathbf{G}$ are the sequences of predicted and ground truth images, respectively, and $i$ indexes the frames. We then flatten these differences to obtain feature vectors for each time step:

$$\mathbf{p} = \text{flatten}(\Delta \mathbf{P}) \tag{12}$$

$$\mathbf{g} = \text{flatten}(\Delta \mathbf{G}) \tag{13}$$

The covariance matrix $\mathbf{C}$ for the flattened vectors is computed as:

$$\mathbf{C} = \frac{1}{N-1} \sum_{i=1}^{N} (\mathbf{p}_i - \bar{\mathbf{p}})(\mathbf{g}_i - \bar{\mathbf{g}})^T \tag{14}$$

where $N$ is the number of time steps, and $\bar{\mathbf{p}}$ and $\bar{\mathbf{g}}$ are the means of the vectors $\mathbf{p}$ and $\mathbf{g}$ respectively. The standard deviation for each vector is calculated as follows:

$$\sigma_{\mathbf{P}} = \sqrt{\text{diag}(\mathbf{C}_{\mathbf{PP}})} \tag{15}$$

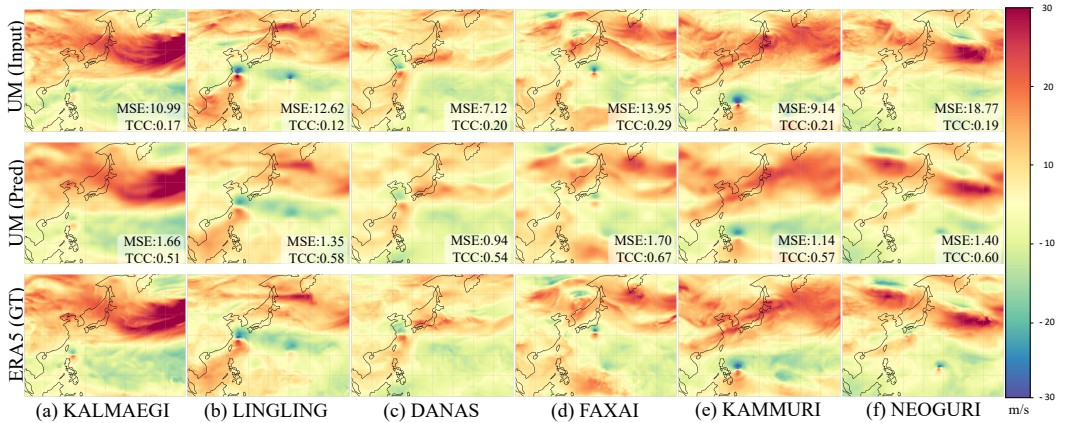

Figure 12: Comparison with zonal wind variables with 700 hPa: From top to bottom, the columns are UM (NWP), bias-corrected UM (output), and ERA5 (ground truth). All MSE and temporal correlation coefficient (TCC) calculations are based on the original physical values.

$$\sigma_{\mathbf{g}} = \sqrt{\mathrm{diag}(\mathbf{C_{gg}})} \qquad (16)$$

where $\mathbf{C_{pp}}$ and $\mathbf{C_{gg}}$ represent the diagonal elements of the covariance matrix corresponding to $\mathbf{p}$ and $\mathbf{g}$. The TCC is then computed as:

$$\mathrm{TCC} = \frac{\mathbf{C_{pg}}}{\sigma_{\mathbf{p}}\sigma_{\mathbf{g}}} \qquad (17)$$

where $\mathbf{C_{pg}}$ is the off-diagonal element of the covariance matrix representing the covariance between $\mathbf{p}$ and $\mathbf{g}$. The final TCC score is the average of these values across all features.

We calculate the TCC for typhoons by cropping a $50 \times 50$ pixel area around the Ground Truth (GT) typhoon locations. Figure 11-(a) presents an example of TCC. As shown in the figure, the correlation coefficient is calculated across all predicted frames. Additionally, as seen in Figure 11-(b), although the bias-corrected UM appears blurred, it accurately represents the center of the typhoon. These results support the high TCC values of the bias-corrected UM, serving as evidence of its effectiveness, as shown in Figure 12

## A.5    MORE DETAILS OF ABLATION STUDY

The ERA5 dataset has been accumulating data from 1950 to the present, whereas the UM dataset has been available since 2010. Therefore, if only the UM dataset is used, the best track typhoon dataset is also only available from 2010. The number of typhoons since 2010 is approximately 200, which is a small fraction compared to the total of 1,424 typhoons that have occurred since 1950. Consequently, the performance of UM-only training is significantly limited. In contrast, a joint training setting, which uses ERA5 data from 1950 to 2009 and the UM dataset from 2010 to 2021, can utilize the entire typhoon dataset. This dataset is about six times larger than UM-only dataset, which seems to significantly improve the performance of data-driven models due to the increased amount of data. Training a physics-conditioned model using only ERA5 and then inferring with UM data seems to result in decreased performance compared to joint training, likely due to the domain gap between ERA5 and UM. However, Bias-correction appears to have improved performance by reducing this domain gap. However, it is evident that joint training has the most significant impact on the performance enhancement. These experimental results indicate that joint training has great potential, especially when ERA5 cannot be used for real-time inference. Furthermore, while we used a physics-conditioned model as a foundation model considering computational resources, future work could also explore methods like domain adaptation.

## A.6    LIMITATION AND FUTURE WORK

Our work has been trained and validated exclusively on typhoons, which are tropical cyclones occurring in the North Western Pacific. Therefore, due to the varying progression directions of

tropical cyclones in different regions, it is impractical to directly apply a model trained in the North Western Pacific to other types or regions of tropical cyclones, even if their physical characteristics are similar. Additionally, our model still relies on NWP models, making it unusable without them. In our future work, we plan to apply LT3P to tropical cyclones occurring globally and provide codes and training recipes for Climate AI. Furthermore, since LT3P is fundamentally a probabilistic prediction model, it requires the selection of a path. In our future work, we plan to conduct research on a selector that chooses a single path, rather than simply relying on ensemble average methods.

## A.7 QUALITATIVE EXPERIMENT RESULTS OF LT3P

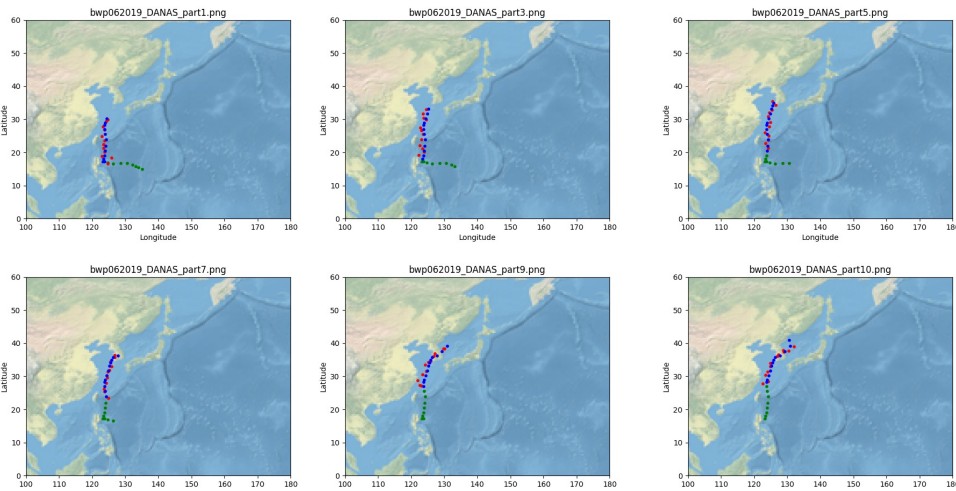

Figure 13: Results showcase trajectory predictions using LT3P for the DANAS Typhoon. The green dots are input, the red dots are prediction, and the blue dots are GT.

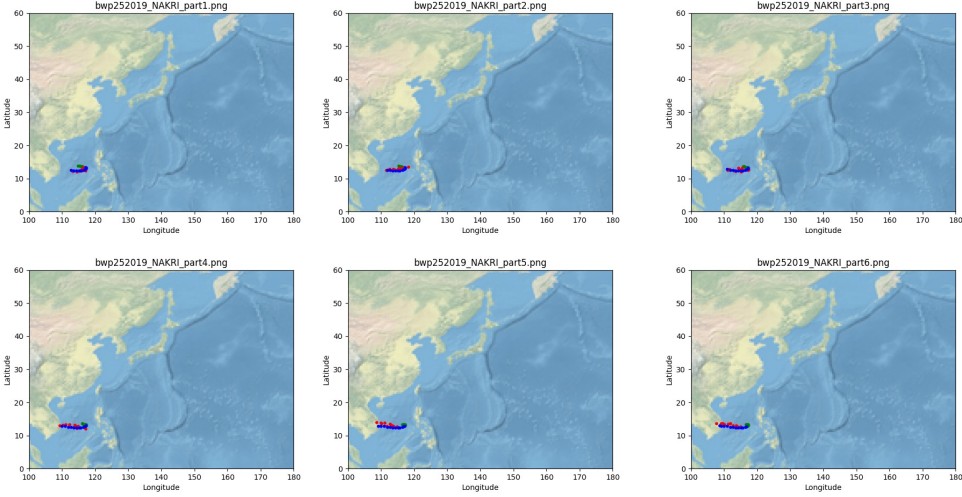

Figure 14: Results showcase trajectory predictions using LT3P for the NAKRI Typhoon. The green dot is input, the red dot is prediction, and the blue dot is GT.

