# OpenReview forum: "Long-Term Typhoon Trajectory Prediction: A Physics-Conditioned Approach Without Reanalysis Data"
_ICLR.cc/2024/Conference — ICLR 2024 spotlight_

### Official Review · Reviewer_cnGt · 2023-10-30

**Soundness:** 3 good
**Presentation:** 3 good
**Contribution:** 3 good
**Rating:** 8
**Confidence:** 4

**Summary:**

This paper proposes a new data-based model for typhoon trajectory prediction, using current and passed typhoon locations as well as unified model (UN) pressure and wind maps.

**Strengths:**

The paper is interesting for both its application results compared to state-of-the-art, and also presents an interesting methodological framework. Indeed, getting rid of reanalysis data (which are not available in real time) is an important asset. The way to do so, first by learning the prediction of the physic variables maps using reanalysis data (which has more history than UN maps, and is also more precise), then to combine using a cross-attention both 'corrected' UN prediction with trajectory prediction, seems to give very good results.
I think this paper would fit at ICLR, yet I have some questions which I would like to know the answer, and also a proofreading should be performed.

**Weaknesses:**

1) The study is limited to one region, as it looks that the UN map has a fixed latitude and longitude values. How was it chosen, what if typhoons are in the borders, or even going outside? It also means that your model can't easily be applied on other regions?

2) Since real-time computation is the goal, it would be important to give computation times values.

3) It is not clear how the 'probability cones' are obtained: stochasticity is mentioned only in the result part, not in the method.

4) I understand that the number of data is limited, but I would like to know if a validation set was used to fine-tune the hyper-parameters or if it was done using the 2019-2021 years. Please explain better.

5) Finally, it would be interesting to see one of the 'worst' cases also in Figure 3, with a comment on it.

6) Many typos are present, see below.

**Questions:**

cf. section 'weaknesses'.

Q1) in your state-of-the art comparison, which one uses UN, which uses only trajectory information?
Q2) how come the Bias-corrected is sometimes better than ERA5 only? Any insights?
Q3) Could you comment a bit more on the Figure 4? it looks like the bias-correction is actually losing the details, just fitting better to the distribution of the ERA5. Maybe there is more to see? The values (SSIM, mean) are not very convincing.
Q4) How many different typhoons are used in the training set?
Q5) Figure 1: what is the scale of the image? how many km (or give the lat/lon) in both directions?

Minor comments/grammar/typos:

- p. 2 'the UM' --> not yet defined (except Abstract)
- 'But, it is not without the drawbacks.' sentence issue
- 'while the UM data has only a 3-hour delay compared to ERA5' --> not clear: is it 3 hour less than ERA5?
- LSTM, MLP, ... not defined acronyms
- 'on a conservation laws'
- 'hyperparameters & a architecture'
- 'The ERA5 dataset has been accumulated at 6-hour intervals from 1950 to the present, and is too large scale to train a model from scratch. It is computationally inefficient to search optimal hyperparameters & a architecture for trajectory predictions.' --> not clear.
- what do you mean by 'this architecture is computationally efficient when training with large-scale ERA5 dataset?
- (6) L_trajectory is not defined.
- 'and use it as our backbone architecture' --> we missing?
- in Evaluation Metrics, MID not defined.
- Fig 3 : we are viewing probability cones? It is not clear
- 'bais-corrected'
- ssim: not defined

---

> ### Author Response · Authors · 2023-11-19
>
> We are pleased that the reviewer acknowledges LT3P has achieved state-of-the-art in the field of typhoon trajectory prediction. We are also thankful for recognizing the effectiveness of the LT3P training pipeline. All of your feedback has been thoroughly reviewed and incorporated into our text. For ease of reference, we've highlighted the updated sections in cyan.
>
> **[W1 - Application to other regions].** The study is limited to one region, as it looks that the UN map has a fixed latitude and longitude values. How was it chosen, what if typhoons are in the borders, or even going outside? It also means that your model can't easily be applied on other regions?
>
> The regions we selected are cropped areas where typhoons, among the numerous types of tropical cyclones, are defined. We have added this information and references on page 7 of the main text. Additionally, in Appendix A.3, we have included inference results as zero-shot validations and analyses for regions defined for other types of tropical cyclones.
>
> **[W2 - Computation time].** Since real-time computation is the goal, it would be important to give computation times values.
>
> All deep learning methodologies can infer within 3 seconds with A100 single gpu. Indeed, additional time is required to obtain input data; however, compared to other methods, it is sufficiently close to real-time. Other models using ERA5 have more than 3 days of data acquisition delays, while our model has only a 3-hour delay. Therefore, it takes about 3 hours to calculate the typhoon's trajectory 72 hours ahead.
>
> **[W3 - Definition of ‘Probability cones’]** It is not clear how the 'probability cones' are obtained: stochasticity is mentioned only in the result part, not in the method.
>
> The probability cones are obtained through kernel density estimation (KDE). We have added this information to Figure 3.
>
> **[W4 - Details of the Dataset].** I understand that the number of data is limited, but I would like to know if a validation set was used to fine-tune the hyper-parameters or if it was done using the 2019-2021 years. Please explain better.
>
> We have included the following content in the main text to address this: The training data spans the years 1950-2018 and includes 1,334 typhoons, while the test dataset covers the years 2019-2021 and comprises 90 typhoons.  Note that for hyperparameter tuning, we train on data from 1950 to 2016 and use the data from 2017 to 2018 as the validation set.
>
> **[W5 - Worst case from our prediction].** Finally, it would be interesting to see one of the 'worst' cases also in Figure 3, with a comment on it.
>
> We have added an analysis of a failure case in Appendix A.4.
>
> **[W6 - Typos].** Many typos are present, see below.
>
> Thank you very much. We have corrected all typos and revised the sentences with a native speaker.

---

> > ### Author Response · Authors · 2023-11-19
> >
> > **[Q1 - SOTA comparisons].** in your state-of-the art comparison, which one uses UN, which uses only trajectory information?
> >
> > Only LT3P uses the UM data. To the best of our knowledge, LT3P is the first work that combines real-time NWP data with ERA5 data. For the use of data type, we add the clarifications for all comparison methods in Table 2.
> >
> > **[Q2 - Our model’s performance compared to ERA5 only].** how come the Bias-corrected is sometimes better than ERA5 only? Any insights?
> >
> > Table 3 shows the results for stochastic models. It presents figures for the best path chosen out of 20 produced paths. When using only ERA5, the LT3P, trained on re-analysis data that implicitly includes typhoon trajectory GT information, focuses solely on the ERA5. In contrast, LT3P trained on UM data cannot accurately predict typhoon trajectories by relying solely on UM data, hence it also utilizes the Best Track data for predictions. Therefore, using Bias-corrected UM predicts a wider distribution covering all areas of feasible typhoon’s future trajectories, which has an advantage when choosing the best one out of 20 paths.
> >
> > **[Q3 - Prediction on Bias-corrected UM].** Could you comment a bit more on Figure 4? it looks like the bias correction is actually losing the details, just fitting better to the distribution of the ERA5. Maybe there is more to see? The values (SSIM, mean) are not very convincing.
> >
> > Based on your comment, we need to provide better analysis by focusing on typhoon trajectory in the time axis. We thus replace the SSIM with the temporal correlation coefficient (TCC), which measures the Pearson correlation coefficient of motion patterns between a predicted and ground-truth trajectory. Detailed analysis can be found in Figure 9&10 and the description of Appendix A.5 section.
> >
> > **[Q4 - Dataset Details].** How many different typhoons are used in the training set?
> >
> > 1,334 typhoons were used in the training set. This information has been reflected on page 7 of the main text.
> >
> > **[Q5 - Dataset Details].** Figure 1: what is the scale of the image? how many km (or give the lat/lon) in both directions?
> >
> > We have added a description on this in the caption of Figure 1. The area covers latitudes from 0 to 59.75°N and longitudes from 100°E to 179.75°E, whose distance is about 59.75 degrees×111 km/degree≈6632.25 km and (179.75−100) degrees×111 km/degree≈8842.75 km, respectively.
> >
> > Thank you for your time reviewing our work and we hope that our comments have helped clarify the matter.

---

### Official Review · Reviewer_y7ao · 2023-10-31

**Soundness:** 3 good
**Presentation:** 2 fair
**Contribution:** 2 fair
**Rating:** 8
**Confidence:** 4

**Summary:**

- This paper proposes a encoder-decoder type weather prediction model, focusing on the typhoon trajectory prediction.
- A notable point is the use of two different reanalysis data. A precise but slow-to-retrieve ERA5 data is used to pretrain the encoder-decoder module. A rough but fast-to-retrieve UM data is then used to fine-tune the encoder-decoder, as well the trajectory predictor.
- Reanalysis data is usually used only for off-line training, However, the UM data is known fir its quick release (3 hours after observations),  one can use the UM for computing the ERA5-like embedding for precise trajectory prediction.
- Enables 72 hours ahead prediction based on a UM reanalysis data.

**Strengths:**

- It is novel to use re-analysis data for a prediction (inference) phase. This can change the possible applications of the reanalysis data, usually used for offline computations only.
- An error correction scheme to utilize the quick but erroneous UM reanalysis effectively sounds interesting.

**Weaknesses:**

- If I read correctly, several explanations about network architecture, losses, ... are missing. This would prevent the re-production by fellow researchers.
  * Number of layers, embedding dimensions, total amounts of learnable parameters, ..
  * L_{trajectory} not defined?

- The reported quantitative results In the Tables are surprisingly good. However, I have the following concerns.
  * Inconsistent trends of the scores of existing (compared) models. As I read the MGTCF paper, the authors reported that the MGTCF performs clearly better than SGAN, which is different from Tables 2, and 3.
  * The scores (Distance) of the proposed LT3P update the current SotA by order of magnitude. I checked the most recent works such as (Bi+ 2023, Lam+ 2022) but the distance scores are roughly in the same order as the existing methods. I feel the current manuscript does not sufficiently explain why this huge jump happens, although the ablation study tells that it seems the joint training with reanalysis is a key factor.

**Questions:**

I'm a little bit confused about how to handle the "Lead time" in evaluations in a fair way.

The manuscript says that one needs to wait for three hours to obtain the UM reanalysis data. This means the proposed method loses the lead time of three hours. So, the score of "6-hour leadtime" should be understood as "3-hour leadtime"?

---

> ### Author Response · Authors · 2023-11-19
>
> We are pleased with the reviewer’s recognition of LT3P’s novelty in Climate AI and their acknowledgment of our pipeline. All comments have been addressed in our revised text, with changes marked in blue.
>
> **[W1 - Missing implementation details]** If I read correctly, several explanations about network architecture, losses, ... are missing. This would prevent the re-production by fellow researchers.
>
> We have utilized three different baselines: GAN, CVAE, and Diffusion. Due to the extensive nature of the contents, we have added detailed explanations in Appendix A.1.
>
> **[W2 - Comparison with MGTCF].** Inconsistent trends of the scores of existing (compared) models. As I read the MGTCF paper, the authors reported that the MGTCF performs clearly better than SGAN, which is different from Tables 2, and 3.
>
> We trained our models using the official repository of MGTCF. In our experiments, SGAN performed better than MGTCF. We suspect this discrepancy may be due to differences in the training and test data used for MGTCF. Although MGTCF utilized ERA5 data, it only trained on cropped areas around the center coordinates of typhoons. We hypothesize that MGTCF might not synergize well with the best track data. In contrast, SGAN, which relies solely on coordinate information, demonstrated better performance than MGTCF.
>
> [https://github.com/Zjut-MultimediaPlus/MGTCF]
>
> **[W2 - Comparison with Sotas].** The scores (Distance) of the proposed LT3P update the current SotA by order of magnitude. I checked the most recent works such as (Bi+ 2023, Lam+ 2022) but the distance scores are roughly in the same order as the existing methods. I feel the current manuscript does not sufficiently explain why this huge jump happens, although the ablation study tells that it seems the joint training with reanalysis is a key factor.
>
> Our LT3P model, which utilizes a 20-sample average ensemble, exhibits performance comparable to deterministic models such as Bi+2023 and Lam+2022, which generate only a single path, as illustrated in Table 2. (For example, Bi+2023 reports a 120km error at a +72-hour prediction.)  Table 3, on the other hand, demonstrates our model's superior performance with the lowest error path selected from 20 inferences, showcasing the benefits of LT3P's stochastic approach.  Consequently, this stochastic ability to infer multiple paths allows us to significantly outperform deterministic models like Bi+2023 and Lam+2022 in Table 3.  Moreover, Bi+2023 and Lam+2022 rely on ERA5 data, which can not infer real-time predictions, and they produce climate variables, not actual typhoon paths, which are subsequently used to identify the typhoon center using a rule-based method. It is important to highlight that LT3P functions as a real-time stochastic model.
>
> **[Q1 - Definition of the “lead time”].** I'm a little bit confused about how to handle the "Lead time" in evaluations in a fair way. The manuscript says that one needs to wait for three hours to obtain the UM reanalysis data. This means the proposed method loses the lead time of three hours. So, the score of "6-hour leadtime" should be understood as "3-hour leadtime"?
>
> Considering a real-world, real-time inference scenario, UM data has a 3-hour delay at data acquisition, the first lead time is +3 hours, and the rest are at +6-hour intervals.  For example, if we assume that we request the UM prediction dataset of 00:00, we can access the data at 03:00. This data represents intervals of UM = {00, +06, +12, +18 … +72}, and though it can be accessed 3 hours later, all the data are at 6-hour intervals. Therefore, while there is a 3-hour delay for inference, from the dataset's perspective, the intervals are consistently every 6 hours
>
> Thank you once again for your feedback. We hope our detailed clarification on the network architecture, model comparison, and significant performance leap of LT3P clearly conveys the novelty of LT3P in Climate AI.

---

> > ### Comment · Reviewer_y7ao · 2023-11-21
> > **Thank you authors for feedback!**
> >
> > Thank you for providing kind answers to my questions.
> > I'm still struggling to read all the review comments and answers. I will post some comments whenever I encounter additional questions and comments.

---

> > > ### Comment · Reviewer_y7ao · 2023-11-23
> > > **Will upgrade my score**
> > >
> > > I feel the author feedbacks are largely reasonable, including responses to other
> > > fellow reviewers.
> > >
> > > It seems the use of two different reanalysis data is fruitful for community. I want the idea is presented and
> > > discussed at ICLR.
> > >
> > > I will upgrade the score to the positive side now, but may be changed again after reviewer discussions.

---

> > > > ### Author Response · Authors · 2023-11-23
> > > >
> > > > Thank you for your review and your points regarding our paper.
> > > >
> > > > We appreciate the opportunity to address these concerns and provide further clarity.
> > > >
> > > > We will release the dataset and code to contribute to the climate AI community.
> > > >
> > > > Thank you again for your kind words and feedback on our work.

---

### Official Review · Reviewer_j264 · 2023-10-31

**Soundness:** 3 good
**Presentation:** 3 good
**Contribution:** 3 good
**Rating:** 6
**Confidence:** 3

**Summary:**

The paper introduces the Long-Term Typhoon Trajectory Prediction (LT3P), a novel approach for real-time typhoon trajectory prediction without the need for reanalysis of data. LT3P is a data-driven model that utilizes a real-time Numerical Weather Prediction (NWP) dataset, making it unique in its field. The model is designed to predict the central coordinates of a typhoon, eliminating the need for additional forecasters and algorithms. LT3P is accessible to various institutions, even those with limited meteorological infrastructure. The paper includes extensive evaluations, demonstrating that LT3P achieves state-of-the-art performance in typhoon trajectory prediction. However, the model is currently applied only to typhoons and has not been tested on other types of tropical cyclones. The authors plan to extend its application in future work and contribute to the field of climate AI by releasing their dataset, training, test codes of LT3P, and pre-trained weights to the public.

**Strengths:**

- Innovative Approach: LT3P is one of the first data-driven models for real-time typhoon trajectory prediction that utilizes a real-time NWP dataset, distinguishing it from other methods in the field.

- Extensive Evaluations: The paper includes comprehensive evaluations, showcasing the model's state-of-the-art performance in typhoon trajectory prediction.

- Contribution to Climate AI: The authors plan to release their dataset, training, test codes of LT3P, and pre-trained weights to the public, contributing valuable resources to the field of climate AI.

**Weaknesses:**

- Limited Application: The model has only been applied to typhoons and has not been tested on other types of tropical cyclones, limiting its current applicability.

- Dependence on Real-Time NWP Dataset: The model's performance is dependent on the availability and accuracy of the real-time NWP dataset, which could be a potential limitation.

- Need for Future Work: While the paper outlines plans for future work, including extending the application to all kinds of tropical cyclones, these aspects have not yet been addressed or tested.

**Questions:**

- How well does the LT3P model generalize to different regions and conditions of typhoon occurrences? Have there been any specific challenges in adapting the model to various geographical locations?

- Could you provide more insight into why LT3P outperforms other state-of-the-art models and established meteorological agencies’ predictions? What specific features or methodologies contribute to its superior performance?

- How does the dependency on real-time NWP datasets affect the model’s performance, especially in scenarios where real-time data might be sparse or inaccurate?

- How does the LT3P model handle uncertainties in typhoon trajectory prediction, and what measures are in place to ensure the reliability of its predictions?

---

> ### Author Response · Authors · 2023-11-19
>
> We sincerely appreciate the reviewer's insightful feedback and acknowledge the highlighted weaknesses. We are truly delighted that the reviewer acknowledges LT3P as one of the first data-driven models for real-time typhoon trajectory prediction. Additionally, their belief in LT3P's value in Climate AI has been a great source of motivation for us. We have addressed all the issues pointed out by the reviewer in our text, marked in olive color.
>
> **[W1 - Application on other types of tropical cyclones].** Limited Application: The model has only been applied to typhoons and has not been tested on other types of tropical cyclones, limiting its current applicability.
>
> From a meteorological perspective, 'typhoons' refer to tropical cyclones that occur in the North Western Pacific region. While the meteorological mechanisms of tropical cyclones may be similar, the movement patterns, such as their trajectories, can differ by region and thus should be treated differently. Additionally, creating a generalized model for global application could lead to computational resource wastage due to the inclusion of excessive irrelevant data from regions not affected by typhoons. Ideally, a single trained model that generalizes well to all tropical cyclones would be beneficial. However, our goal has been to achieve higher accuracy by training models specifically for each region.
>
> Nevertheless, as seen from the experimental results in Appendix A.3, our LT3P shows the potential for application in diverse regions. We perform the zero-shot validation of our model for cyclones in both the North Indian Ocean and the South Indian Ocean. We note that the prediction results from their meteorological agencies are not available. As expected, our method outperforms the comparison methods. However, the FDE is larger than the test results in the North Western Pacific.
>
> **[W2 - Dependence on NWP Dataset].** Dependence on Real-Time NWP Dataset: The model's performance is dependent on the availability and accuracy of the real-time NWP dataset, which could be a potential limitation.
>
> Our model focuses on leveraging the advantages of real-time NWP datasets, and therefore, it does not function properly in situations where NWP datasets are entirely unavailable. As the reviewer point out, this can be seen as a potential limitation of our study. However, our model requires only three basic types of meteorological variables, which in most cases can be easily obtained from freely available outputs of NWP models. A prime example is the use of the publicly available dataset from ECMWF's IFS forecasts. Known for their high accuracy, these IFS forecasts are a reliable source. To account for the practical scenario of using NWP model outputs with varying accuracies, we opt for the UM forecasts, known to have a larger bias. This decision is made to demonstrate that our model can yield impressive results even with NWP datasets of somewhat lower accuracy.
>
> **[W3 - Future work].** Need for Future Work: While the paper outlines plans for future work, including extending the application to all kinds of tropical cyclones, these aspects have not yet been addressed or tested.
>
> In our future work, we will validate LT3P on all types of tropical cyclones occurring globally. We have added future work in Appendix A.7.

---

> > ### Author Response · Authors · 2023-11-19
> >
> > **[Q1 - Applications on different regions (different types of tropical cyclones)].** How well does the LT3P model generalize to different regions and conditions of typhoon occurrences? Have there been any specific challenges in adapting the model to various geographical locations?
> >
> > We have applied our LT3P model to tropical cyclones occurring in different regions. Appendix A.3 shows the results of applying LT3P to different areas. In conclusion, LT3P does not suffer significant performance degradation because it uses the UM prediction, a global model without a domain gap. However, additional training appears necessary for optimal performance.
> >
> > **[Q2 - LT3P Performance].** Could you provide more insight into why LT3P outperforms other state-of-the-art models and established meteorological agencies’ predictions? What specific features or methodologies contribute to its superior performance?
> >
> > The key idea of LT3P is to incorporate NWP dataset into typhoon coordinates. However, our model does not solely rely on NWP which is somewhat erroneous like other methods. LT3P   leverages the rich representation of reanalysis data to take full advantage of the foundation model.
> >
> > **[Q3 - Dependence on NWP dataset ].** How does the dependency on real-time NWP datasets affect the model’s performance, especially in scenarios where real-time data might be sparse or inaccurate?
> >
> > Our current work essentially requires NWP datasets. Therefore, it cannot be utilized when access to NWP data is not possible. However, since we have chosen UM among many NWPs, it is expected to perform well with other higher-performing NWPs like IFS and GFS.
> >
> > **[Q4 - Uncertainty and reliability of prediction].** How does the LT3P model handle uncertainties in typhoon trajectory prediction, and what measures are in place to ensure the reliability of its predictions?
> >
> > Currently, we minimize the uncertainties through multi-modal prediction which produces a set of outputs from one observation trajectory. The multi-modal trajectories are averaged for better probabilistic forecasting of one dominant path, which is conventionally used in meteorology. As another aspect, we can represent the uncertainty, which is visualized in Figure 3. As the probability map of typhoons, we show the coverage of potential pathways using kernel density estimation. And, we do not have measures for the reliability of the predictions. We believe that there is an optimal way to select one dominant path from them, which has been described in our future work. Please refer to Appendix A.7.
> >
> > Thank you once again for your feedback. We hope our detailed discussion on LT3P's regional applications and limitations clearly conveys our dedication to advancing Climate AI.

---

### Official Review · Reviewer_k3qU · 2023-11-01

**Soundness:** 2 fair
**Presentation:** 1 poor
**Contribution:** 2 fair
**Rating:** 3
**Confidence:** 4

**Summary:**

This paper introduces a framework for data-driven typhoon prediction based on cross-attention between encoded representations of the past trajectory points as well as global weather data (geopotential and wind directions). The paper shows good results in forecasting the typhoon trajectories, especially compared to physics-based baselines.

**Strengths:**

Overall I believe that the paper has great potential for presenting an interesting case study with strong empirical results on this important problem, but in its current form the lack of clarity and details for better understanding the study are severe issues.

- The results seem strong and quite significant gains over physics-based baselines (coming from various meteorological institutions) are reported. I am not familiar with the typhoon track forecasting literature though.
- Open-sourcing the data will be valuable. When doing so, the authors should put care into making it easily accessible and well-documented
- Training on NWP data such as UM instead of ERA5 is a good study to make but has limited originality.

**Weaknesses:**

- Clarity can be improved significantly. Firstly, there are a lot of grammar mistakes worth fixing with a grammar checker. Secondly, many implementation details are conveyed very unclearly or details are missing. This inhibits understanding the significance of the paper and would hurt reproducibility. For example:
    - The method seems to have been tested using a GAN, a CVAE, and a diffusion model, but implementation details are largely lacking
    - Unclear meaning and implementation details of *"so it is matched through the latitude and longitude mapping and the interpolation method"*
    - Unclear what exactly the *"ensemble method"* refers to and how exactly it is performed in practice for all the models.
    - Please include more ablations, e.g. with UM and joint training and bias-correction but without pre-training. The whole ablation study section is a bit hard to understand and should be expanded (in the appendix if needed).
- Are the baselines in the first set of rows of Table 2 (e.g SocialGAN) all off-the-shelf pre-trained human trajectory prediction models? If so, it is not surprising at all that they perform so badly, and it would be good to retrain them on your data. Additionally, why is Ruttgers et al. not included in the benchmarking?

**Questions:**

- First sentence of methods section: 1) There is no index *i* for none of the variables (e.g. it should be $c_i$, I think); 2) $t_p$ is the input sequence **length** (not the input sequence). Similarly for $t_f$; 3) $p$ is given different meanings for $C_p$ and the pressure levels $p \in P$. This is confusing, can you use different letters, please?
- Why are geopotential height and wind vectors inputs taken from the output timesteps $i=t_p+1, \dots, t_p+t_f$? Should it not be $i=1, \dots, t_p$?
- 3.1 Preliminaries (not preliminarily)
- Tone down "is too large scale to train a model from scratch". There exist models trained from scratch on ERA5...
- "$\tilde X$ is the forecasting values for ERA5 from time $t$ to $t+t_f$". Shouldn't it be $t+1$ to $t+t_f$?
- I don't completely understand how the bias correction phase works, is the UM data matched with ERA5 based on the timestamp?

---

> ### Author Response · Authors · 2023-11-19
>
> We thank the reviewer for the insightful feedback concerning clarity and details. Your comment makes our paper polish up. Based on comments, we have revised it, marked in orange. Below, we address each of your questions with corresponding answers.
>
> **[S1 - Dataset Release]**. Open-sourcing the data will be valuable. When doing so, the authors should put care into making it easily accessible and well-documented.
>
> Thank you for acknowledging our Physics Track dataset. As shown in Appendix A.2, we have already prepared the project page interface and details for public release. Currently, due to the large size of our dataset, we are unable to upload an anonymized link. We plan to make the dataset publicly available upon the acceptance of our paper.
>
> **[W1 - Implementation Details]** The method seems to have been tested using a GAN, a CVAE, and a diffusion model, but implementation details are largely lacking.
>
> We have included detailed information about the implementation in Appendix A.1 section to provide clarity for readers. In particular, we have mainly described a loss function, embedding and component  of each baseline.
>
> **[W2 - Detail of implementation]** Unclear meaning and implementation details of "so it is matched through the latitude and longitude mapping and the interpolation method.
>
> The original meaning of the sentence is that we just utilize a bilinear interpolation to adjust a resolution of UM data and ERA5. Instead of the expression that the reviewer point out, we have changed it to “Note that the resolution has been adjusted using bi-linear interpolation”. Please check it on page 7 of the main text.
>
> **[W3 - Definition of ‘ensemble method’]** Unclear what exactly the "ensemble method" refers to and how exactly it is performed in practice for all the models.
>
> We have employed the ensemble average method for comparison, similar to the approach used by meteorological offices. Since each inference from the GAN, CVAE, and Diffusion models produces different outputs due to random noise, we average the generated 20 trajectory samples from our model for better probabilistic forecasting. On page 7 of the main text, this method is clearly referred to as the ‘ensemble average '.
>
> **[W4 - More experiments and details on ablation study].** Please include more ablations, e.g. with UM and joint training and bias-correction but without pre-training. The whole ablation study section is a bit hard to understand and should be expanded (in the appendix if needed).
>
> Unfortunately, due to the inherent characteristics of our model structure, conducting bias-correction independently without the pre-training phase is not feasible.
> The ERA5 dataset has been accumulating data from 1950 to the present, whereas the UM dataset has been available since 2010. Therefore, if only the UM dataset is used, the best track typhoon dataset is also only available from 2010. The number of typhoons since 2010 is approximately 200, which is a small fraction compared to the total of 1,424 typhoons that have occurred since 1950. Consequently, the performance of UM-only training is significantly limited. In contrast, a joint training setting, which uses ERA5 data from 1950 to 2009 and the UM dataset from 2010 to 2021, can utilize the entire typhoon dataset.This dataset is about six times larger than UM-only dataset, which seems to significantly improve the performance of data-driven models due to the increased amount of data
> Lastly, we have added more content on the analysis of ablation studies in Appendix A.6 section.
>
> **[W5 - Implementation details].** Are the baselines in the first set of rows of Table 2 (e.g SocialGAN) all off-the-shelf pre-trained human trajectory prediction models? If so, it is not surprising at all that they perform so badly, and it would be good to retrain them on your data.
>
> We note all baselines in Table 2 are trained and evaluated using the best track dataset for strictly fair comparison. To avoid misunderstanding, we have specified the training dataset of the baselines on page 7.
>
> **[W5 - Selection of comparison models].** Additionally, why is Ruttgers et al. not included in the benchmarking?
>
> We did not include Ruttgers et al.'s work in our benchmarking efforts due to differences in both the training dataset and methodology. Ruttgers et al. track typhoon paths by reconstructing them after marking the typhoon center coordinates on satellite images. In contrast, we do not use satellite images.
>
> However, it would be valuable to conduct an experiment based on their work. We have now requested the authors for their codes and datasets. Should these sources become available, we will carry out an additional comparison with Ruttgers et al.'s work, and include the result in our paper.

---

> ### Author Response · Authors · 2023-11-19
>
> **[Q1 - Notation Issues].** Confusing notation.
>
> Thank you for pointing out our mistakes. We have specified the index 'i' on page 4 and corrected the duplicated notation 'p.' To avoid confusion, we have changed all the small ‘p’ to ‘o’ which represents observations in descriptions and figures. “”
>
> **[Q2 - Timestamps of UM and ERA5].** Why are geopotential height and wind vectors inputs taken from the output timesteps.
>
> The output time steps represent the intervals at which we aim to predict the typhoon paths, and leveraging atmospheric information for these time frames is crucial for accurate prediction. This goal is achieved by utilizing the Numerical Weather Prediction (NWP) forecast fields, especially the UM dataset.
> We use physics-based predicted values from UM as inputs, offering a significant advantage over using ERA5. Unlike ERA5, which not only experiences delays but also lacks access to future data, the UM forecast incorporates future atmospheric conditions through physics-based modeling. This makes it a more suitable choice for our input values.
>
> **[Q3 - Grammar error]** . 3.1 Preliminaries (not preliminarily)
>
> As the reviewer pointed out, we have corrected the typo. Do you think any further modifications are needed for the content of this section? We believe explaining how our dataset is derived from the NWP model is essential. Therefore, we have composed this section to convey that information.
>
>
> **[Q4 - Tone].** Tone down "is too large scale to train a model from scratch". There exist models trained from scratch on ERA5…
>
> We have toned down the sentence as “The ERA5 dataset, accumulated at 6-hour intervals from 1950 to the present, requires a significant amount of time to train a model from scratch. To address this challenge, we have adopted a strategy of constructing a foundational model for typhoon trajectory prediction”. This is reflected in Section 3.2 on page 5.
>
> **[Q5. - Timestamp of bias-correction output]** Timestamp of \tilde{X}
>
> The output time frame of \( \tilde{x} \) is from \( t_{o}+1 \) to \( t_{o} + t_{f} \).
>
> **[Q6 - Timestamp of bias-correction output].** I don't completely understand how the bias correction phase works, is the UM data matched with ERA5 based on the timestamp?
>
> Yes, you are correct. In our bias correction phase, we train the model on ERA5 data for the period from 1950 to 2010, ensuring it outputs ERA5 data. Simultaneously, for the period from 2010 to 2021, UM data is used as an input, and the bias correction guides the model to produce ERA5-like output from UM data.
>
> Thank you once again for reviewing our work. We hope our comments have addressed any questions and concerns and clarified things. We look forward to your further feedback on our revised work.

---

### Author Response · Authors · 2023-11-19

We deeply respect the dedication of all reviewers to the academic community. To efficiently address the reviewers' questions, we have written our rebuttal in a Q & A format. Below are the abbreviations we have used:

Strengths -> S
Weaknesses -> W
Questions -> Q

Thank you!

---

### Public Comment · ~JiayangWu1 · 2025-04-30

Dear Authors,

Thank you for your impressive work. We are currently reproducing your results for fair comparison but have encountered several issues.

First, the released code appears incomplete, particularly in terms of module integration. For example, the `TrajectoryTransformer` model invokes components like `IAM4VP`, `CrossAttention`, and the encoder/decoder blocks, but their connections are loosely defined or absent in the shared code, making it difficult to reconstruct a runnable pipeline end-to-end.

Second, while the paper states that ERA5 variables (geopotential height, u-wind, v-wind at 250, 500, and 700 hPa) are normalized and formatted as (B × 12 × 9 × 240 × 320), the preprocessing scripts used to convert raw ERA5 data into the expected `.npy` format are not provided. In addition, the number of processed samples we obtain does not align with the paper (1,334 train / 90 test).

The total ERA5 data size we estimated is under 1.5 TB — which seems manageable enough to be shared via Google Drive.

If possible, we would greatly appreciate it if you could share the missing preprocessing code or provide access to the preprocessed ERA5 dataset to support faithful and fair reproduction of your results.

Best regards,

---

### Meta-Review · Area_Chair_PZ3j · 2023-12-10

**Metareview:**

This paper introduces the Long-Term Typhoon Trajectory Prediction (LT3P) that allows to leverage results coming from Numerical Weather Prediction (NWP) in a causal dependency, without the need for reanalysis of the data using data from the future, and not necessitating high-quality reanalysis data like ERA5 at runtime but being able to use coarser UM data instead. The model is trained on reanalysis ERA5 data then transferred onto real-time UM predictions, and it can predict the central coordinates of a typhoon. The model is evaluated on typhoon data only. Data, training and test code as well as model weights will be open-sourced.


Strengths:
* Strong, state-of-the-art results and significant gains over physics-based baselines (k3qU,j264,cnGt)
* Open-sourced data, training and test code (k3qU,j264) and contribution to climate AI.
* Training on real-time NWP data is novel (k3qU,j264,y7ao,cnGt)
* Extensive evaluation (j264)
* Interesting error correction of UM data (y7ao)

Weaknesses:
* k3qU, y7ao and cnGt pointed out lack of clarity in the paper, including implementation details: authors addressed these in the revision.
* k3qU requested additional ablations with bias correction: the authors replied this was not possible due to the recency of some climate datasets.
* k3qU noted that the SocialGAN was not retrained from human trajectory to typhoon trajectory prediction: the authors corrected the misunderstanding.
* Limited evaluation beyond typhoons (i.e., no other tropical cyclones) (j264, cnGt): the authors had actually evaluated the method in the Indian ocean as well, and added further evaluation in the rebuttal phase.
* y7ao had questions about some MGTCF baseline and about the reason for improvement over the SOTA, addressed by the authors.
* cnGt requested for computation time values.

Addressing some points raised by the reviewers:
* I do not understand the point made by k3qU: "Training on NWP data such as UM instead of ERA5 is a good study to make but has limited originality."
* Reviewer j264 said that "The model's performance is dependent on the availability and accuracy of the real-time NWP dataset, which could be a potential limitation." but such problem cannot reasonably be addressed by the weather and climate prediction community, and ECMWF does provide free and global Integrated Forecast Service data.

Reviewer k3qU did not respond to the authors and did not engage in the discussion; moreover, there is a discrepancy between the text of the review (which would correspond to a score 4 or 5, not 3) and thus I have recalibrated their score from 3 to 5.

The authors have addressed all reviews in a satisfactory way. The authors have also made extensive correction and rewriting to the paper, using colour-code responses for each reviewer, to clearly indicate their changes.

Based on the scores (3->5, 6, 8, 8) and reviews, I vote for accepting this paper.

**Justification For Why Not Higher Score:**

The strength of the paper is in its execution on climate data to solve a specific problem (typhoon trajectory prediction) due to the slow nature of reanalysis, more than on a general problem for the wider ICLR community.

**Justification For Why Not Lower Score:**

This paper suffered from harsh evaluation by k3qU and the authors have made extensive changes to the paper during rebuttal time, adding new baselines and details, and have addressed all points raised by the reviewers. Reviewers who read the rebuttal raised scores to 8.

---

### Decision · Program_Chairs · 2024-01-16

Accept (spotlight)